# The ACC-Deaminase Producing Bacterium *Variovorax* sp. CT7.15 as a Tool for Improving *Calicotome villosa* Nodulation and Growth in Arid Regions of Tunisia

**DOI:** 10.3390/microorganisms8040541

**Published:** 2020-04-09

**Authors:** Khouloud Bessadok, Salvadora Navarro-Torre, Eloísa Pajuelo, Enrique Mateos-Naranjo, Susana Redondo-Gómez, Miguel Ángel Caviedes, Amira Fterich, Mohamed Mars, Ignacio D. Rodríguez-Llorente

**Affiliations:** 1Valuation of Biodiversity and Bioresources in Dry Lands (BVBZA), Faculty of Sciences of Gabès Zrig, 6029 Gabès, Tunisia; khouloudbessadok@gmail.com (K.B.); ftirich_amira@yahoo.fr (A.F.); Mohamed.Mars@fsg.rnu.tn (M.M.); 2Departamento de Microbiología y Parasitología, Facultad de Farmacia, Universidad de Sevilla, 41012 Sevilla, Spain; snavarro1@us.es (S.N.-T.); epajuelo@us.es (E.P.); caviedes@us.es (M.Á.C.); 3Departamento de Biología Vegetal y Ecología, Facultad de Biología, Universidad de Sevilla, 41080 Sevilla, Spain; emana@us.es (E.M.-N.); susana@us.es (S.R.-G.)

**Keywords:** arid soils, ACC deaminase, legumes, nodule associated bacteria, rhizobia

## Abstract

*Calicotome villosa* is a spontaneous Mediterranean legume that can be a good candidate as pioneer plants to limit regression of vegetation cover and loss of biodiversity in Tunisian arid soils. In order to grow legumes in such soils, pairing rhizobia and nodule associated bacteria (NAB) might provide numerous advantages. In this work, cultivable biodiversity of rhizobial symbionts and NAB in nodules of *C. villosa* plants growing in five arid regions of south Tunisia was characterized. Phylogenetic analysis using 16S rDNA gene, *dnak*, *recA* and *nodD* sequences separated nodule-forming bacteria in six clades associated to genera *Ensifer*, *Neorhizobium*, *Phyllobacterium* and *Rhizobium*. Among NAB, the strain *Variovorax* sp. CT7.15 was selected due to its capacity to solubilise phosphate and, more interestingly, its high level of aminocyclopropane-1-carboxylate deaminase (ACC deaminase) activity. *C. villosa* plants were inoculated with representative rhizobia of each phylogenetic group and co-inoculated with the same rhizobia and strain CT7.15. Compared with single rhizobia inoculation, co-inoculation significantly improved plant growth and nodulation, ameliorated plant physiological state and increased nitrogen content in the plants, independently of the rhizobia used. These results support the benefits of pairing rhizobia and selected NAB to promote legume growth in arid or degraded soils.

## 1. Introduction

Soils in Tunisia’s arid regions are characterized by low nutrient content, which is often a limiting factor in plant production. Hence, these areas suffer from rapid regression of vegetation cover, loss of biodiversity and a significant decline in biological activity [1]. Therefore, to limit the massive genetic erosion of species, it is necessary to promote the cultivation of spontaneous plants, especially those used for the restoration of nitrogen soil imbalance. *Calicotome* belongs to the *Genisteae* tribe of the subfamily *Faboideae*; this genus includes four species: *Calicotome spinosa*, *Calicotome villosa*, *Calicotome infesta* and *Calicotome intermedia* [2]. *Calicotome villosa* is a spontaneous Mediterranean legume (a 30–150 cm spiny shrub displaying yellow flowers during the spring) that is quite common in the north of Africa and Spain [3]. *Calicotome* can be pioneer plants growing in barren soils due to a deep and wide root system and their capacity, as legume, to fix atmospheric nitrogen in symbiosis with rhizobia. Rhizobia are Gram-negative soil bacteria classified within α- and β- proteobacteria subclasses that induce the development of root nodules in legumes. Inside the nodule, rhizobia reduce atmospheric N into ammonia to be assimilated by the plant in exchange for plant-derived organic acids [4]. *Calicotome* spp. nodule-forming bacteria are frequently slow-growing rhizobia affiliated to *Bradyrhizobium* [5,6]. Contrary to these results, fast-growing rhizobia were identified among isolates from root nodules of *C. villosa*, which were affiliated to *Rhizobium* genus [7,8]. 

An increasing number of studies on rhizobial diversity using standard cultivation methods have reported the presence of the so-called nonrhizobial endophytes (NRE) [9] or nodule associated bacteria (NAB) [10] in nodules of different legume species [11]. Besides rhizobia, other rhizospheric bacteria colonize the interior tissues of plants and contribute in favourable ways to the plant development. These bacteria belong to different genera including *Acinetobacter*, *Agrobacterium*, *Arthrobacter*, *Bacillus*, *Bosea*, *Enterobacter*, *Micromonospora*, *Mycobacterium*, *Paenibacillus*, *Pseudomonas*, *Stenotrophomonas* and *Variovorax*, among others [11,12]. Even though those bacteria do not induce nodule formation, they might be able to colonize the nodules formed by rhizobia strains, and this association may be beneficial for both partners [11,13,14]. These nodule endophytes can behave as plant-growth-promoting bacteria (PGPBs), enhancing plant growth by a variety of direct and indirect mechanisms [15]. Direct mechanisms include facilitation of nutrient acquisition, such as atmospheric nitrogen fixation; production of siderophores for iron uptake and phosphate solubilization; production of phytohormones, like auxins, abscisic acid or cytokinins; and the presence of 1-aminocyclopropane-1-carboxylate deaminase (ACC deaminase) activity, among others [15,16]. Another direct mechanism to increase legume nodulation can be the creation of additional infection sites [17]. PGPB inhabiting nodules and legume rhizosphere involved in plant growth through facilitating plant nodulation could be named as rhizobia helper bacteria (RHB) [17,18].

The aim of this work was to examine the cultivable biodiversity of nodule-inducing rhizobia and NAB in nodules of *C. villosa* plants growing in arid regions of south Tunisia and also to select NAB that could facilitate plant nodulation and growth, in order to establish rhizobia–NAB couples with potential use in the revegetation of these arid soils.

## 2. Materials and Methods 

### 2.1. Localization of the Plants and Characteristics of the Soils

*Calicotome villosa* nodules were collected from wild plants growing in five arid zones of Tunisia: Gabès, Matmata, Zarate, Médenine and Tataouine (Appendix A, Table 1). pH of the soil was determined using a portable meter and a calibrated electrode system (CrisonpH/mVp-506, Hach Lange SLU, Barcelona, Spain). The conductivity of the soil water was determined using a conductivity meter (Crison-522, Hach Lange SLU, Barcelona, Spain) after dilution with distilled water (1:1). Lastly, three 0.5 g dry subsamples of each soil were digested with 6 mL HNO_3_ (3:1, *v*/*v*), 0.5 mL HF and 1 mL H_2_O_2_ at 130 °C for 5 h, and ion concentrations were measured by inductively coupled plasma (ICP) spectroscopy (ARL-Fison3410, Thermo Scientific, Waltham MA, USA).

### 2.2. Isolation and Bacterial Growth

Bacterial strains were isolated from nodules of wild *C. villosa* plants growing in arid soils described in the previous section. Root nodules were dissected from roots and rinsed thoroughly in water. Surfaces were first disinfected for 1 min with 95% ethanol followed by sodium hypochlorite (5% (*v*/*v*)) for 3 min and then extensively rinsed several times with sterile distilled water. Nodules were then crushed on sterile plates and streaked onto yeast-extract mannitol agar (YMA). Plates were incubated at 28 °C for 3–5 days [19]. The purity of the culture was validated by picking and restreaking individual colonies on fresh plates. Bacteria with different colony morphology, among those isolated from nodules of plants growing in the same location, were selected for further identification. To assess the efficiency of the disinfection procedure, for each isolation experiment two sterile and noncrushed nodules were rolled over YMA agar and incubated under the same conditions as the samples. Bacteria were also grown in YMA plates at increasing temperatures ranging from 25 to 40 °C, pH from 4 to 9 and NaCl concentrations from 1% to 2.5% for 3–5 days. 

### 2.3. Genetic Diversity by BOX-PCR and Identification of Cultivable Bacteria

Genomic DNA was extracted using the G-spin genomic DNA extraction kit (INtRon Biotechnology, Inc., Gyeongii-do, South Korea), following the manufacturer’s instructions. Genetic diversity in bacteria isolated from nodules collected in each region was studied by BOX-PCR using BOX A1R primer (5′-CTA CGG CAA GGC GAC GCT GAC G-3′). PCR steps were programmed as described in Mesa et al. [20] using 40 ng of genomic DNAs as templates. PCR products were electrophoresed through a 1.5% agarose gel at 75 V for 2 h. Gel was visualized under UV radiation and photographed. The gel was photographed, and the fingerprints were analysed with CLIQS 1D Pro Software (TotalLab Ltd., Newcastle-Upon-Tyne, UK). Dendrograms were reconstructed using the BOX-PCR profile. The similarities were determined by calculating Pearson’s product moment correlation coefficient [21].

Bacteria representative of each genetic profile were identified by PCR amplification and sequencing of the 16S rDNA following the conditions described in Rivas et al. [22]. EzTaxon server was used to determine homologies with DNA sequences from described bacterial type strains [23]. Partial 16S rDNA sequences were deposited in GenBank (Table 2).

Housekeeping genes *recA* (recombinase A) and *dnak* (chaperone protein DnaK) were amplified using primer couples recA6F/recA555R and dnak1466F/dnak1777R following the method described by Mertens et al. [24]. PCR amplification of symbiotic *nodD* gene was performed following the conditions described in [22,25]. 

PCR products were purified with ExoSAP-IT PCR Product Cleanup (Thermo Fisher, Waltham, MA, USA), and sequencing was performed at StabVida (Lisbon, Portugal). For phylogenetic analysis, sequences were assembled and then aligned using ClustalW program [26], with relevant sequences obtained from the GenBank database. The phylogenetic trees with 1000 bootstrap replications were constructed using MEGA 6.0 software [27]. Selected gene sequences obtained in this work were deposited in the GenBank database, and their accession numbers are shown in the phylogenetic trees.

### 2.4. Determination of PGP Properties and Enzymatic Activities

Bacterial cultures were plated on nitrogen-free broth (NFB) medium as preliminary test for nitrogen fixation [28]. Strains able to grow in NFB medium were tested for acetylene reduction following the protocol described in [29]. Positive phosphate solubilization was resolved on NBRIP (National Botanical Research Institute’s phosphate growth medium) plates when bacterial growth caused the appearance of surrounding transparent halos [30]. Plates were incubated 3 days at 28 °C. The quantification of IAA (indole-3-acetic acid) was evaluated by a colorimetric method as detailed in Mesa et al. [20]. Quantitative measurement of ACC deaminase activity of bacterial strains was carried out using the method described by Penrose and Glick [31]. The reaction was determined by comparing the absorbance at 540 nm of the sample to a standard curve of α-ketobutyrate. Then, total protein concentration of toluenized cells was estimated using bovine serum albumin (BSA) to create the protein calibration curve [32]. Finally, the enzyme activity was calculated based on the µmoles of released α-ketobutyrate per mg of protein per hour.

Lipase and protease activities were performed growing the strains in Tween80 and casein agar plates, respectively, and the presence of halos around bacteria growth indicates positive enzymatic activities [33]. Chitinase activity was observed according to Mesa et al. [20]. Pectinase and cellulase activities were examined as described in Elbeltagy et al. [34]. DNAse activity was studied in DNA agar plates and revealed with 1 N HCl. Concerning amylase activity; bacteria were grown in starch agar (Scharlab SL, Barcelona, Spain) plates. Plates were revealed by covering plate surface with 10 mL of lugol.

### 2.5. Plant Inoculation and Growth

Seeds of *Calicotome villosa* were first treated with pure sulfuric acid for 1 min, rinsed abundantly six times with sterile distilled water, and then allowed to germinate on 1% agar plates at 28 °C for approximately 72 h. Germinated seeds were transferred to sterilized plastic pots containing sterile soil from Gabès region. Soil was sterilized at 121 °C, 1 atm for 20 min, three times (mixing vigorously the soil after each sterilization cycle). Seeds were irrigated with nitrogen-free buffered nodulation medium (BNM) [35] and inoculated with 10^8^ bacterial cells per mL of the desired bacterial inoculum. For each strain, three replicates were considered. Noninoculated plants were included as negative controls. Plants were rinsed with BNM approximately once a week for 5 months and nodulation was observed. Cross-inoculation assays using *Medicago sativa*, *Lens culinaris* and *Lotus edulis* were developed using the same protocol except for seeds germination. In that case, seeds were treated with 5% sodium hypochlorite and rinsed six times with sterile distilled water.

For co-inoculation experiments, germinated seeds of *C. villosa* were inoculated with the selected isolates by immersion in 10^8^ cells per mL solution (resuspended in BNM), transferred into plastic pots filled with sterilized soil from Gabès and then inoculated with the remaining bacteria. The experimental design consisted of two replicates (three plants per replicate) of each of the four following treatments: (i) noninoculated control plants; (ii) plants inoculated with the ACC-deaminase-producing strain *Variovorax* sp. CT7.15; (iii) plants inoculated separately with selected nodule-inducing rhizobial strains: *Phyllobacterium* sp. CTEM17, *Rhizobium* sp. CZ2.7, *Rhizobium pakistanense* CE36, *Neorhizobium* sp. CZ4.17 and *Ensifer* sp. CM61; and (iv) plants co-inoculated with the selected rhizobia and the strain *Variovorax* sp. CT7.15. Pots were placed in the greenhouse (18/8 h light/dark at 25/23 °C) and rinsed with BNM solution once a week for 5 months. Plants were then collected and data regarding growth parameters, i.e., plant height, shoot and root dry weight and number of nodules, were recorded. 

### 2.6. Photosynthetic Pigments

At the end of the experimental period, photosynthetic pigments in fully developed green leaves from each treatment were extracted using 0.05 g of fresh material in 10 mL of 80% aqueous acetone. After filtering, 1 mL of the suspension was diluted with a further 2 mL of acetone, and chlorophyll a (Chl a) and chlorophyll b (Chl b) contents were determined with a HitachiU-2001 spectrophotometer (Hitachi Ltd., Tokyo, Japan) using three wavelengths (663.2, 646.8 and 470.0 nm). Concentrations of pigments (µg × g of fresh weight^−1^) were obtained following the method described by Lichtenthaler [36].

### 2.7. Statistical Analysis

Statistical analysis was performed using the software Statistica v. 6 (Statsoft Inc., Tulsa, OK, USA). Data were first tested for normality with the Kolmogorov–Smirnov test and for homogeneity of variance with the Brown–Forsythe test. Comparisons between means in different inoculation and co-inoculation treatments at the end of the experiment were made by using generalizer linear models (GLM). Significant test results were followed by Fisher tests (LSD) for identification of important contrasts. 

## 3. Results and Discussion

### 3.1. Biodiversity of Cultivable Bacteria Inhabiting C. villosa Nodules in Tunisian Arid Soils

Since limited information was available on bacteria nodulating or NAB in *C. villosa*, the first objective of this work was therefore to identify and characterize the nodule cultivable bacteria associated with this legume in arid soils of Tunisia, particularly in five regions with similar chemical properties, low organic material and phosphate content (Table 1).

Based on place where nodules were collected and colony morphology, 43 cultivable bacterial strains were originally isolated. Nodulation tests using *C. villosa* plants were performed to evaluate the ability of the isolates to renodulate de host plant. Twenty-seven isolates were able to nodulate *C. villosa* plants under controlled conditions, 12 failed to nodulate and 4 induced small nonfixing white lumps. Isolates able to renodulate *C. villosa* plants were classified as fast growing (able to grow in 3–5 days at 28 °C). 

BOX-PCR fingerprinting of the isolates allowed grouping by genetic profiles (Appendix A). Here, 16S rRNA of 21 strains were sequenced and compared with type strains using Eztaxon server [23] (Table 2). The selection of these strains was performed as follows: among strains with the same BOX-PCR profile only one of them was sequenced (i.e., CZ61 and CZ7.21 or CE5.11 and CE4.6 showed the same profile in Appendix A, and only CZ7.21 and CE5.11 were sequenced). In the same way, strains with undefined profile (i.e., CE36 in Appendix A) were also sequenced. Our results showed the existence of a wide biodiversity in *C. villosa* nodules, with isolates belonging to 10 different genera. 

Strains were also characterized based on some phenotypic parameters, such as growth at different NaCl concentrations, increasing pH and temperature, as well as antibiotic tolerance Although isolates showed a wide diversity in their characteristics, most of them were able to grow at pH ranging from 5 to 9 and temperatures ranging from 25 to 35 °C. In general, isolates showed a low tolerance to NaCl, and only strain CM61 related to *E. meliloti* grew at 1% NaCl.

### 3.2. Rhizobia Isolated from C. villosa Nodules in Tunisian Arid Soils

Concerning rhizobia, strains related to *Phyllobacterium* sp. were isolated from nodules of *C. villosa* plants growing in Gabès, Médenine and Tataouine regions and *Rhizobium* sp. in Zarate, Médenine and Tataouine. In Matmata region, nodules were elicited by *Ensifer* sp. strains. *Neorhizobium* sp. strains were isolated from nodules in Zarate, in addition to strains related to *Rhizobium pakistanense*, which were also isolated in Médenine. These strains nodulated *C. villosa* in greenhouse conditions. Finally, strains that induced small white lumps in *C. villosa* roots were related to *Rhizobium radiobacter* and isolated from plants growing in Gabès soils.

The phylogenetic relationship between the sequenced strains relative to type strains based on 16S rRNA gene grouped rhizobia into six clades (Figure 1). Strains CZ4.12 and CZ4.17 were closely related to *Neorhizobium alkalisoli* species. Isolates CE36 and CZ7.21 were closely related to *Rhizobium pakistanense*. Three strains, CT4.4, CE 5.13 and CZ2.7, were related to *Rhizobium sullae* and *Rhizobium esperanzae*. CM61 was related to *Ensifer meliloti*, and CM8.1, also identified as *Ensifer* sp., was not clearly associated with any species. CE5.11, CT17 and CTEM17 were identified as *Phyllobacterium* sp. and related to *Phyllobacterium ifriqiyense* and *Phyllobacterium catacumbae*. Finally, CG2.6 and CG2.8 were identified as *Rhizobium radiobacter*. 

The housekeeping genes are considered as robust markers for assessing the evolutionary genetics of rhizobia [37]. Thus, to confirm the phylogenetic position of the strains isolated in this study, *recA* and *dnak* genes of one strain of each clade were sequenced. Strain CT4.4, positioned between *R. sullae* and *R. esperanzae*, was also included to clarify its position. Consistent with the 16S rRNA tree, the concatenated phylogenetic tree based on *recA* and *dnak* sequences grouped *C. villosa* nodulating isolates in six different clades (Figure 2). CE5.13 and CT4.4 were closely related to *Rhizobium sullae*. CG2.6 and CZ7.21 had concatenated sequences that were identical to *R. radiobacter* and *R. pakistanense*, respectively, with a high bootstrap support (100%). With lower bootstrap support, strain CM61 was clustered with *Ensifer meliloti* LMG 6133^T^. The other strains were distributed between two clusters: strain CZ4.17 aligned with *Neorhizobium galegae* CCBAU 01393^T^ and closely related with *N. alkalisoli*, and CTEM17 affiliated with *Phyllobacterium* cluster with high identity values.

Considering these results and previous works [7,38,39], *Rhizobium* and *Ensifer* seem to be the main microsymbionts for indigenous legumes in Tunisia. Concerning *Rhizobium*, our isolates were closely related to *R. sullae*, which was known to nodulate *Hedysarum coronarium* L. [40], and to *R. pakistanense*, recovered from groundnut nodules in Pakistan [41]. *Ensifer* strains were related to *E. meliloti*, a common microsymbiont of most wild legumes in Tunisian arid soils [39,42,43]. Isolates belonging to *Phyllobacterium* have been previously described to nodulate wild legumes in Tunisia [7,38,43]. *Phyllobacterium* strains isolated in this work were closely related to *Phyllobacterium ifriqiyense* STM370. The first isolation of this reference strain was reported in [44], from nodules of *Astragalus algerianus* and *Lathyrus numidicus* in the infra-arid zone of southern Tunisia. On the other hand, the presence of *Agrobacterium tumefaciens* (formerly *Rhizobium radiobacter*) in nodules of legumes growing in Tunisia was previously reported [39,43,45,46].

None of our isolates was affiliated to *Bradyrhizobium*, unlike in previous studies on *C. spinosa* in the northeast of Algeria [6] and *C. spinosa* and *C. infesta* in different areas of Sicily [5]. Several authors have demonstrated that *Bradyrhizobium* predominates among rhizobia nodulating *Genisteae* [47,48,49,50]. However, in some other studies, Genistoid plants, such as *Genista saharae*, *Retama raetam* or *Retama sphaerocarpa*, were nodulated by *Sinorhizobium* (formerly *Ensifer*), *Rhizobium*, *Mesorhizobium* and *Phyllobacterium* [39,51,52].

The bacterial nodulation (*nod*) genes, which encode the production of Nod factor determine the *Rhizobium*-legume symbiosis [53]. The *NodD* (LysR-type regulator), which acts as a transcriptional activator for the *nod* operon, plays an important role in the control of rhizobial host specificity [54]. In order to determine the nature of the symbiotic interaction and the diversity of *C. villosa* isolates, the symbiotic *nodD* genes were analysed (Figure 3). Strain CE5.13 *nodD* sequence displayed a high similarity to that of *Ensifer kummerowiae* CCBAU71714^T^. This type strain, isolated from *Kummerowia stipulacea*, nodulated its host plant and *Medicago sativa*, but not several other legumes assayed [55]. On the other hand, the strain CM61 shared a 100% *nodD* sequence similarity with *Ensifer kostiense* HAMBI1489^T^, which nodulated *Acacia senegal* and *Prosopis chilensis* [56]. Strains CT4.4, CZ2.7 and CZ7.21, grouped closely together and formed a separate linage (supported by high bootstrap values (97%)) from all other strains, that was closest to *E. medicae*, *E. meliloti* and *E. kummerowiae*. We could not amplify *nodD* gene in any of the *Neorhizobium* sp., *R. radiobacter* or *Phyllobacterium* sp. strains.

Amplification of *nodA* gene, more commonly used for symbiotic characterization of the strains, was also attempted using different primer couples as described by Tounsi-Hammami et al. [57]. Unfortunately, *nodA* gene could not be amplified after several attempts. Accordingly, Tounsi-Hammani et al. [57] were able to amplify *nodA* gene only in 1 out of 14 selected rhizobia able to nodulate *Lupinus albus* in Tunisian calcareous soils.

The symbiotic genes of rhizobia are reported to have an evolutionary history different from those of the housekeeping genes [58], an argument that is supported by the *nodD*, 16S and housekeeping gene phylogenies obtained in this study. This may be explained by the fact that the symbiotic genes that codify nodulation are plasmid borne in fast-growing rhizobia, so they can be transferred among species in the rhizosphere.

Finally, a cross-inoculation experiment to test the ability of these autochthonous rhizobia to nodulate representative grain legumes was developed (Appendix A). *R*. *sullae* and *E*. *meliloti* strains nodulated *M. sativa*, *L. culinaris* and *L. edulis*. *N. galegea* and *R. pakistanense* strains only nodulated *M. sativa*, and *Phyllobacterium* strains did not nodulate any of the legumes assayed. 

### 3.3. NAB Isolated from C. villosa Nodules in Tunisian Arid Soils

The presence of nodule associated bacteria (or NAB) could be preferentially selected by legumes as these bacteria could provide benefits to counteract nutritional deficiencies, drought and other stresses. Among NAB isolated, *Brevundimonas* sp. and *Sphingomonas* sp. strains accompanied *Phyllobacterium* sp. in nodules collected in Gàbes. Nodules induced by *Ensifer* sp. in Matmata contained *Sphingomonas* sp. and *Pseudomonas* sp. strains. *Bacillus* sp. was found inhabiting nodules collected in Zarate. Finally, NAB isolated from nodules of Tataouine belonged to genera *Inquilinus*, *Pseudomonas* and *Variovorax*. Based on 16S rRNA gene sequences, a phylogenetic tree of cultivable NAB was constructed (Figure 4).

Zakhia et al. [8] reported the presence of *Bacillus*, *Inquilinus* and *Sphingomonas* in nodules of spontaneous *C. villosa* plants growing in Tunisia. In the same work, *Pseudomonas* strains were found in nodules of other spontaneous legumes growing in the same region. *Variovorax* has been isolated from nodules of *Acacia* growing in southeastern Australia [59] and nodules of *Crotalaria* growing in Ethiopia [60].

### 3.4. PGP Properties of the Isolates

Assessing the presence of PGP properties is an important task, enabling the selection of the most efficient bacterial inoculants. In that way, the abilities to solubilize phosphate and to produce IAA and the presence of ACC deaminase activity were assayed in all the isolates. Most of the nodule-forming rhizobacteria (29 out of 31) were able to solubilize phosphate, while only the *Variovorax* sp. strains among the nonrhizobia did it. Soil phosphate is frequently found in insoluble form in soil, so bacteria with phosphate-solubilising activity may provide an available form to the plant. Concerning IAA production, a hormone directly involved in plant root elongation, only the strain *R. radiobacter* CG2.6 was able to produce low hormone levels (0.84 mgL^−1^). Strains *Brevundimonas* sp. CTEM2.9 and *Variovorax* sp. CT7.5 and CT7.15 showed ACC deaminase activity. Particularly, *Variovorax* sp. CT7.15 showed a high level of activity, up to 27.26 µmol α-ketobutyrate mg of protein^−1^h^−1^. The phytohormone ethylene (C_2_H_4_) is an important modulator of plant growth and developmental processes [61,62]. For many plants, C_2_H_4_ stimulates germination and breaks the seed dormancy, but, during germination, a high concentration of ethylene inhibits root elongation [63]. Bacteria having ACC deaminase activity can regulate ethylene levels in the plant, thereby limiting the damage to the plant [31].

Nitrogen fixation was also studied in NAB isolated from *C. villosa* nodules. Only strains of *Variovorax* sp., both CT7.5 and CT7.15, grew in nitrogen-free medium. Nevertheless, they could not reduce acetylene, indicating that they did not fix atmospheric nitrogen. 

Based on these results, *Variovorax* sp. CT7.15 was selected to study its capability to improve *C. villosa* growth and nodulation in co-inoculation with nodule-inducing rhizobia.

The presence of enzymatic activities related with organic matter degradation in *Variovorax* sp. CT7.15 was also studied. This strain showed pectinase, amylase and lipase activities; and was negative for cellulase, chitinase, DNAse and protease. 

### 3.5. Co-Inoculation of C. villosa with Rhizobia and Variovorax sp. CT7.15 Strain Improves Plant Nodulation and Growth 

The need for more research on pairing rhizobia and NAB in order to grow legumes on arid or eroded soils has been recently pointed out [11]. In this context, surface disinfected *C. villosa* seeds were inoculated with selected rhizobia or co-inoculated with the same rhizobia and the strain *Variovorax* sp. CT7.15. One nodule-forming rhizobia of each phylogenetic clade, among those that induced the formation a greater number of nodules in re-inoculation experiments (data not shown), was selected.

As expected, inoculation with rhizobia increased plant growth (Figure 5). Inoculation improved both plant height and dry weight, particularly due to an increase in shoot weight, independently of the rhizobia (Figure 5A,C). Nevertheless, significant differences in aerial part dry weight were only found in plants inoculated with *Rhizobium* sp. CZ2.7 (Figure 5A). On the other hand, plants co-inoculated with rhizobia and *Variovorax* sp. CT7.15 showed significantly higher shoot dry weight (Figure 5A), root dry weight (Figure 5B) and plant height than those inoculated only with rhizobia (Figure 5C). Compared to single inoculation, the increases in shoot biomass ranged from 30% (*R. pakistanense*) to 100% (*Neorhizobium* sp.) (Figure 5A), and increases in plant height were around 30% (Figure 5C). Concerning root dry weights, co-inoculated plants duplicated the root weight of plants inoculated only with rhizobia (Figure 5B). Increase in root biomass could be very helpful in the absorption of water and nutrients in arid soils.

The effect of plant inoculation either with rhizobia or co-inoculation with CT7.15 on chlorophyll concentrations was also recorded (Figure 6). Inoculation with rhizobia increased chlorophyll a and b concentrations compared to control plants, with their values being significantly higher in plants inoculated with *Phyllobacterium* sp. and *Rhizobium* sp. Plants co-inoculated with rhizobia and strain CT7.15 showed higher concentrations of both chlorophylls than plants inoculated only with rhizobia, with significant differences independent of the rhizobia (Figure 6A,B). These results indicated that co-inoculation with the *Variovorax* strain CT7.15 ameliorates the physiological state of the *C. villosa* plants. 

Finally, nodulation and plant N content were also evaluated in *C. villosa* plants inoculated with rhizobia and co-inoculated with the *Variovorax* sp. strain (Figure 7). *C. villosa* plants co-inoculated with rhizobia and the ACC-deaminase-producing bacterium showed higher number of nodules than plants inoculated only with rhizobia (Figure 7A). Differences in nodule number were significant in plants inoculated with *Rhizobium* sp. CZ2.7 and *Neorhizobium* sp. CZ4.17 strains (Figure 7A). This effect on nodulation rate might be explained by the ACC-deaminase activity of CT7.15, which decreases levels of ethylene, a well-known negative regulator of nodulation [64,65].

As expected, plants inoculated with rhizobia showed higher values of N than control plants, with statistically significant differences (Figure 7B). On the other hand, co-inoculated plants showed higher values in N content than plants inoculated only with rhizobia, with significant differences independent of the rhizobia (Figure 7B). Although plants inoculated with CT7.15 strain showed higher values of N than control plant, differences were not significant (Figure 7B). Increase in N content in co-inoculated plants might be related not only with the increase in nodule number but also with the ACC deaminase activity of CT7.15, since this enzyme converts ACC (precursor of ethylene) into ammonia and α-ketobutyrate, providing the plant a nitrogen source [66].

## 4. Conclusions

In conclusion, co-inoculation of *C. villosa* plants with autochthonous nodule-inducing rhizobia and the strain *Variovorax* sp. CT7.15 promoted plant growth and nodulation and improved plant physiological state in arid soils, increasing the effects of single rhizobia inoculation. These results support the benefits of pairing rhizobia and selected NAB to grow legumes in arid or degraded soils. 

## Figures and Tables

**Figure 1 microorganisms-08-00541-f001:**
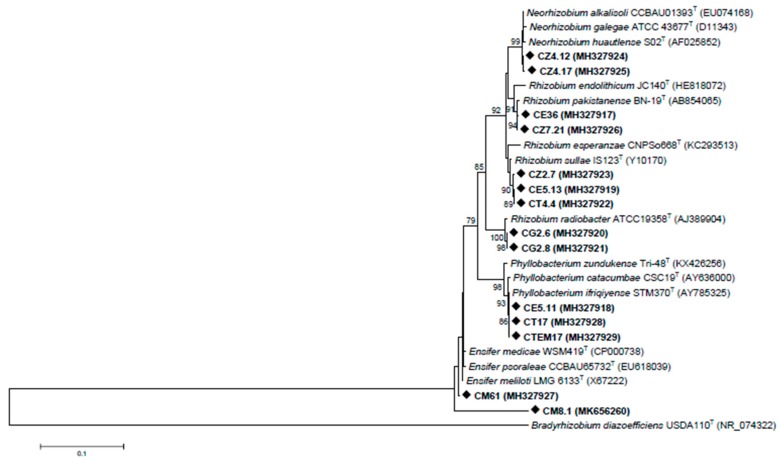
Neighbour-joining phylogenetic tree based on the 16S rRNA gene of representative nodule-inducing strains isolated from nodules of *Calicotome villosa*. Bootstrap confidence levels were derived from 1000 replications, and those greater than 75% are indicated at the internodes. Scale bar = 0.1 nucleotide divergence.

**Figure 2 microorganisms-08-00541-f002:**
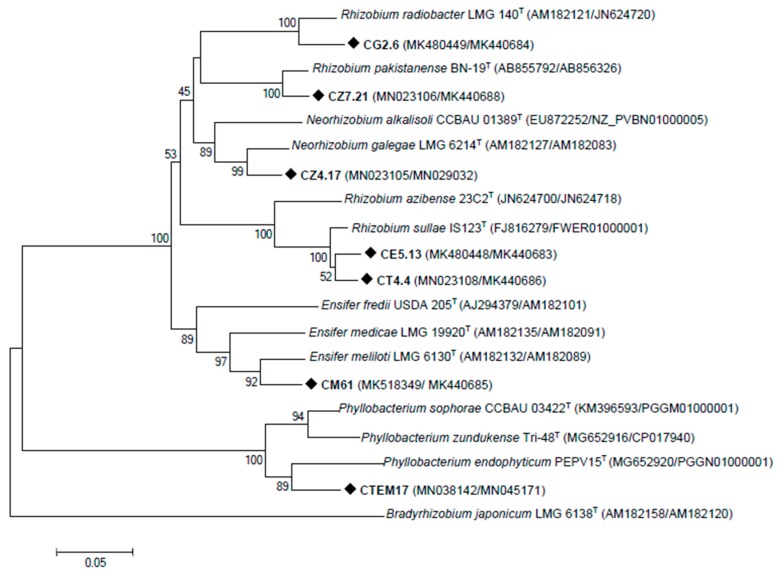
Neighbour-joining tree based on concatenated *recA* and *dnaK* gene sequences of representative nodule-inducing strains isolated from nodules of *Calicotome villosa*. The significance of each branch is indicated by the bootstrap value calculated for 1000 replicates (only values higher than 75% are indicated). Scale bar = 0.02 nucleotide divergence.

**Figure 3 microorganisms-08-00541-f003:**
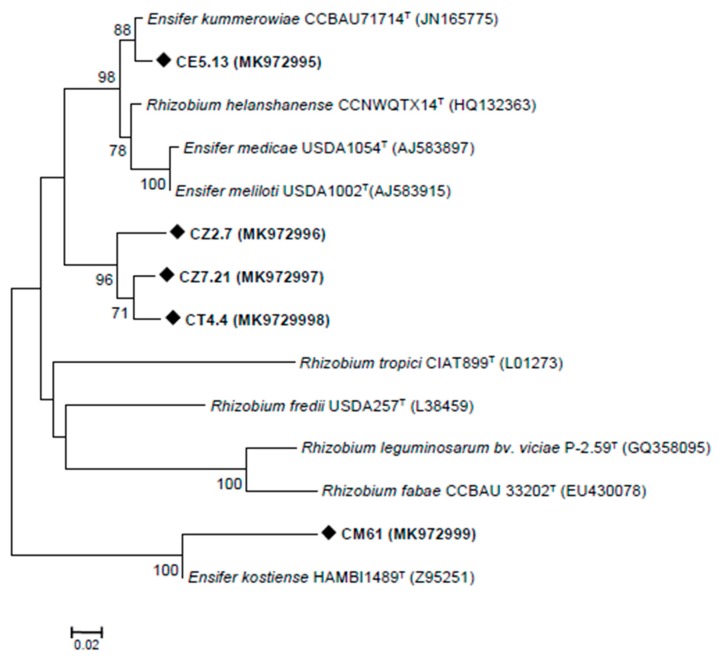
Molecular phylogenetic analysis using the neighbour-joining method of *C. villosa* root nodule isolates for the *nodD* gene. Significant bootstraps (70%) are indicated as percentages (1000 replications). Scale bar = 0.02 nucleotide divergence.

**Figure 4 microorganisms-08-00541-f004:**
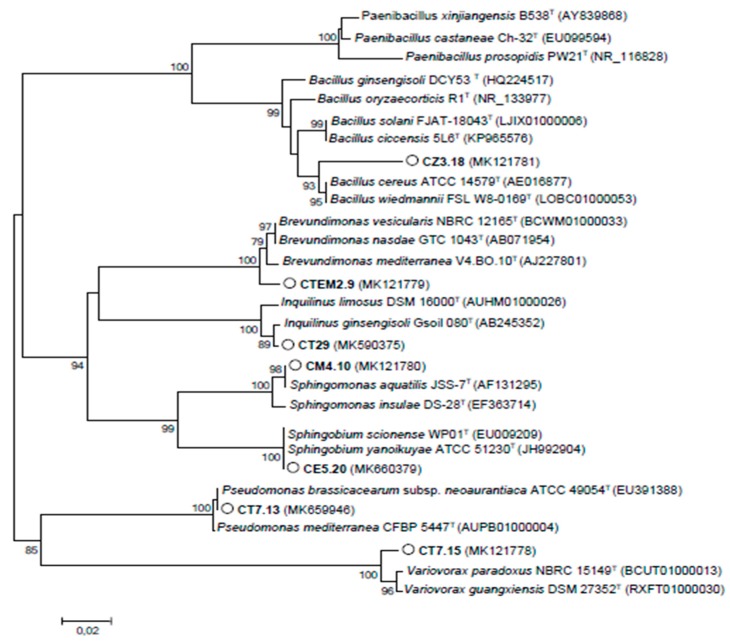
Neighbour-joining phylogenetic tree based on the 16S rRNA gene of endophytic strains isolated from nodules of *Calicotome villosa*. Bootstrap confidence levels were derived from 1000 replications, and those greater than 75% are indicated at the internodes. Scale bar = 0.1 nucleotide divergence.

**Figure 5 microorganisms-08-00541-f005:**
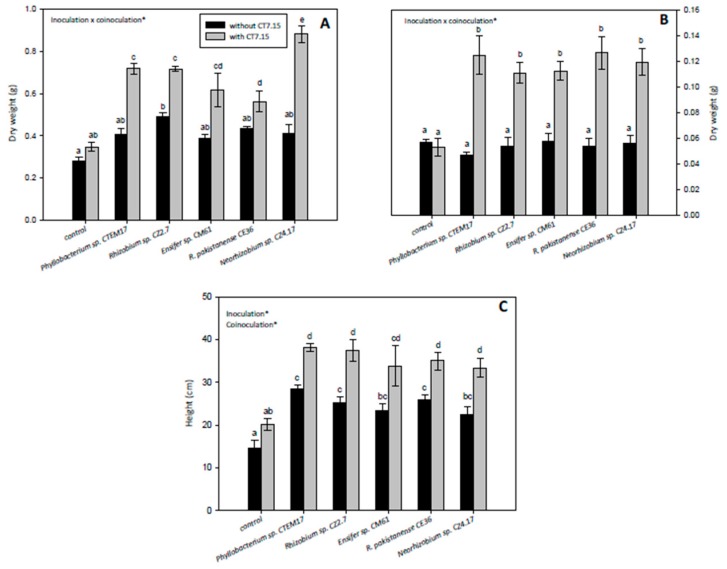
Aerial part dry weight (**A**), root dry weight (**B**) and plant height (**C**) of *Calicotome villosa* in response to inoculation with different rhizobia (*Phyllobacterium* sp. CTEM17, *Rhizobium* sp. CZ2.7, *Ensifer* sp. CM61, *R. pakistanense* CE36 and *Neorhizobium* sp. CZ4.17) in absence and presence of strain CT7.15 as co-inoculant for 5 months. Values represent mean ± SE, *n* = 6. Different letters indicate means that are significantly different from each other (GLM, inoculation × co-inoculation; LSD test, *p* < 0.05). Inoculation, co-inoculation or inoculation × co-inoculation in the corner of the panels indicate main or interaction significant effects (∗ *p* < 0.001).

**Figure 6 microorganisms-08-00541-f006:**
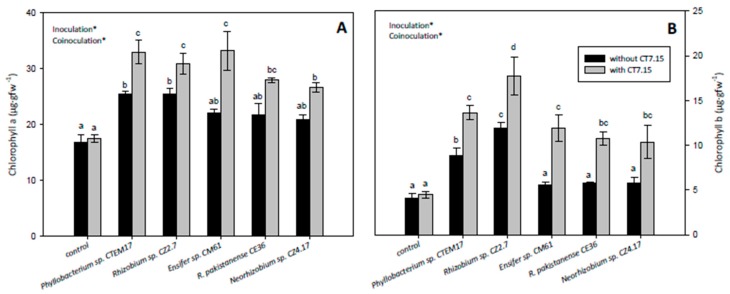
Physiological parameters. Chlorophyll a (**A**) and chlorophyll b (**B**) in randomly selected leaves of *Calicotome villosa* in response to inoculation with different rhizobia (*Phyllobacterium* sp. CTEM17, *Rhizobium* sp. CZ2.7, *Ensifer* sp. CM61, *R. pakistanense* CE36 and *Neorhizobium* sp. CZ4.17) in absence and presence of strain CT7.15 as co-inoculant for 5 months. Values represent mean ± SE, *n* = 6. Different letters indicate means that are significantly different from each other (GLM, inoculation × co-inoculation; LSD test, *p* < 0.05). Inoculation, co-inoculation or inoculation × co-inoculation in the corner of the panels indicate main or interaction significant effects (∗ *p* < 0.001).

**Figure 7 microorganisms-08-00541-f007:**
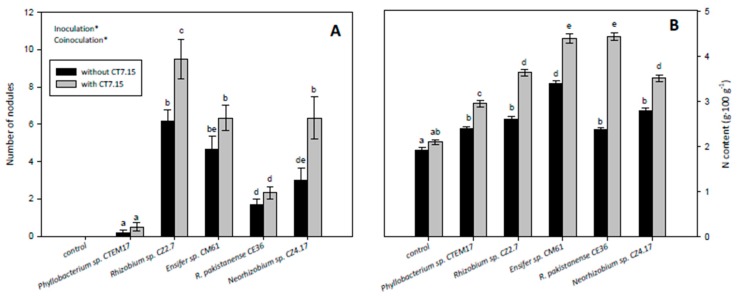
Number of nodules (**A**) in *Calicotome villosa* and nitrogen content (**B**) in randomly selected leaves of *Calicotome villosa* in response to inoculation with different rhizobia (*Phyllobacterium* sp. CTEM17, *Rhizobium* sp. CZ2.7, *Ensifer* sp. CM61, *R. pakistanense* CE36 and *Neorhizobium* sp. CZ4.17) in absence and presence of strain CT7.15 as co-inoculant for 5 months. Values represent mean ± SE, *n* = 6. Different letters indicate means that are significantly different from each other (GLM, inoculation × co-inoculation; LSD test, *p* < 0.05). Inoculation, co-inoculation or inoculation × co-inoculation in the corner of the panels indicate main or interaction significant effects (∗ *p* < 0.001).

**Table 1 microorganisms-08-00541-t001:** Origins of *Calicotome villosa* nodules isolates, pH and conductivity of soils and ecological characteristics, C and P content of their sampling sites.

	Coordinates	Ecosystem and Soil Characteristics	pH	Conductivity (µS/cm^2^)	Corg %	P_2_O_5_ %
S1—Gabes	33°46′ N, 10°5′ E	Oued bed, sandy gravel	8.2 ± 0.3	19 ± 0.5	0.16 ± 0.01	<0.005
S2—Matmata	33°29′ N, 10°4′ E	Oued bed, sandy gravel	9.1 ± 0.4	48 ± 0.4	0.34 ± 0.02	<0.005
S3—Zarate	33°41′ N, 10°21′ E	Sandy (protected area)	8.3 ± 0.2	20 ± 0.6	0.11 ± 0.01	<0.005
S4—Medenine	33°14′ N, 10°19′ E	Sandy gravel	9.3 ± 0.2	32 ± 0.3	0.16 ± 0.03	<0.005
S5—Tataouine	33°11′ N, 10°22′ E	Sandy gravel	8.4 ± 0.3	29 ± 0.4	0.36 ± 0.02	<0.005

**Table 2 microorganisms-08-00541-t002:** Closest species to the representative strains isolated from *Calicotome villosa* based on the 16S rRNA partial sequence using EzTaxon server [23].

Strain	Accession No.	Related Species	Sequenced Fragment (bp)	Identity (%)
CE36	MH327917	*Rhizobium pakistanense*	928	98.99
CE5.11	MH327918	*Phyllobacterium ifriqiyense*	1029	99.70
CE5.13	MH327919	*Rhizobium sullae*	957	99.34
CG2.6	MH327920	*Rhizobium radiobacter*	1303	99.54
CG2.8	MH327921	*Rhizobium radiobacter*	1336	98.39
CT4.4	MH327922	*Rhizobium esperanzae*	1381	98.22
CZ2.7	MH327923	*Rhizobium esperanzae*	1360	97.41
CZ4.12	MH327924	*Neorhizobium alkalisoli*	1300	99.77
CZ4.17	MH327925	*Neorhizobium alkalisoli*	925	99.67
CZ7.21	MH327926	*Rhizobium pakistanense*	1017	98.80
CM61	MH327927	*Ensifer meliloti*	1020	99.51
CT17	MH327928	*Phyllobacterium catacumbae*	1432	95.36
CTEM17	MH327929	*Phyllobacterium ifriqiyense*	1382	96.44
CT17.15	MK121778	*Variovorax paradoxus*	1328	96.61
CTEM29	MK121779	*Brevundimonas* sp.	1100	97.36
CM4.10	MK121780	*Sphingomonas aquatilis*	1302	97.93
CZ3.18	MK121781	*Bacillus cereus*	889	97.64
CT7.13	MK659946	*Pseudomonas* sp.	979	96.78
CM8.1	MK656260	*Ensifer meliloti*	1049	90.14
CE5.20	MK660379	*Sphingomonas* sp.	1229	97.15
CT29	MK590375	*Inquilinus* sp.	1347	97.99

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
