# Peer review of "The ACC-Deaminase Producing Bacterium *Variovorax* sp. CT7.15 as a Tool for Improving *Calicotome villosa* Nodulation and Growth in Arid Regions of Tunisia"

_microorganisms, 2020, doi:10.3390/microorganisms8040541_

Round 1
Reviewer 1 Report
The manuscript describes studies on different (symbiotic and non-symbiotic) strains isolated from Calicotome villosa nodules, and their potential use for enhancing growth of plant host.
The novelty of this work is moderate, because lot of similar papers was published up till now. On the other hand, this specific plant-bacterial model was not studied, and obtained results may lead to development of formulation of new bacterial biofertilizer, which can be applied in defined environment (alkaline sandy soil).
I think the paper is worth of publishing, but it needs pretty large correction. Some sections (like 1. Introduction, 2. Materials and methods, and 3.5 Co-inoculation of C.villosa…) are well-written and easy to understand. On the contrary, some sections of Results (sections 3.1, 3,2, 3.3) are chaotic and unconsistent, and they need to be re-written.
In lines 203-207 the Authors wrote that they isolated 43 strains, i.e. (a) 27 which were able to nodulate plant host, (b) 4 forming non-fixing nodules, and (c) 12 non-nodulating. And this should be a starting point: 27 strains vs. 16 strains, for selection of symbiont (using 27 strains) and selection of helper strain (using 16 strains), which should be described separately.
Detailed comments:
- Selection of strains: lines 208-210: The authors write: “16S rRNA of one representative strain of each profile (…) were sequenced…”. What does “representative strain” mean? Dendrogram on Fig S2 shows that each strain has different BOX-PCR profile, so all of them should be sequenced. Consequently, if the Authors study only selected sequences, they should not be able to write that non-sequenced strains are “related” to any reference strain (Table 3). This can be written only after comparing of sequences.
I think some strains could be arbitrarily chosen (like those listed in Table 2 – please move this table from Materials and Methods into Results section), but the explanation of this choice written at l. 208-210 is not good, please provide another one. And delete Tab. 3.
- Dendrograms
In this paper the Authors provide few dendrograms, but each of them contains different number of studied strains. I can agree that the Authors have chosen 21 strains from whole collection, but this set of strains should be consequently studied. Unfortunately, Fig 1 dendrogram contains 14 sequences of studied strains, Fig 2 – 7 sequences and Fig 3 – only 5 sequences. Please provide 16S, dnaK, recA and nodD sequences for all 14 strains listed in Fig.1, and use them to supplement Fig 2 and Fig. 3. After this, Fig 1 will be unnecessary.
In Results section also the reason for choosing rhizobial strains used in plant tests should be better explained.
- Supplementary materials
Table S1 may be deleted – please move data concerning soil C and P content into Table 1. In my opinion Al and Fe content is not connected with the focus of the study.
Also Table S2 does not contain useful information which might affect the process of selection of strains. Please delete.
Table S4 is very large and superfluous, because most of strains does not differ in their characterisctics. Please delete – most important information from this table can be written in 2-3 sentences.
Author Response
Please, find our response in the attached document.

Reviewer 2 Report
In the current study, the authors reported the ACC-deaminase producing bacterium Variovorax sp. CT7.15 as a tool for improving Calicotome villosa nodulattion and growth in arid regions of Tunisia. The study is well organized and well presented. Also the findings are of interest.
Author Response
We have revised the manuscript for minor spell check. Thank you very much for your time to review our work.